# A Survey of PAPR Techniques Based on Machine Learning

**DOI:** 10.3390/s24061918

**Published:** 2024-03-16

**Authors:** Bianca S. de C. da Silva, Victoria D. P. Souto, Richard D. Souza, Luciano L. Mendes

**Affiliations:** 1National Institute of Telecommunications, Santa Rita do Sapucaí 37540-000, Brazil; victoria.souto@inatel.br (V.D.P.S.); lucianol@inatel.br (L.L.M.); 2Department of Electrical and Electronic Engineering, Federal University of Santa Catarina, Florianópolis 88040-900, Brazil; richard.demo@ufsc.br

**Keywords:** 6G networks, PAPR reduction, artificial intelligence, machine learning

## Abstract

Orthogonal Frequency Division Multiplexing (OFDM) is the modulation technology used in Fourth Generation (4G) and Fifth Generation (5G) wireless communication systems, and it will likely be essential to Sixth Generation (6G) wireless communication systems. However, OFDM introduces a high Peak to Average Power Ratio (PAPR) in the time domain due to constructive interference among multiple subcarriers, increasing the complexity and cost of the amplifiers and, consequently, the cost and complexity of 6G networks. Therefore, the development of new solutions to reduce the PAPR in OFDM systems is crucial to 6G networks. The application of Machine Learning (ML) has emerged as a promising avenue for tackling PAPR issues. Along this line, this paper presents a comprehensive review of PAPR optimization techniques with a focus on ML approaches. From this survey, it becomes clear that ML solutions offer customized optimization, effective search space navigation, and real-time adaptability. In light of the demands of evolving 6G networks, integration of ML is a necessity to propel advancements and meet increasing prerequisites. This integration not only presents possibilities for PAPR reduction but also calls for continued exploration to harness its potential and ensure efficient and reliable communication within 6G networks.

## 1. Introduction

Over the last few decades, mobile communication networks have undergone constant evolution. In the 1980s, a significant milestone in global telecommunications was reached with the emergence of First Generation (1G) mobile communication networks [1]. During this period, various 1G standards were developed around the world. The United States adopted Advanced Mobile Phone System (AMPS) as a prominent standard for 1G, employing Frequency Division Multiple Access (FDMA) to provide voice services [1,2,3,4]. Internationally, different 1G standards gained recognition. The Nordic Mobile Telephone (NMT) system, originating in the Nordic countries and operating on the 450 MHz frequency band, became notable [5,6]. The United Kingdom and some other regions embraced the Total Access Communication System (TACS), operating at around 900 MHz [7]. Japan implemented the C-450 standard, an analog cellular system on the 450 MHz band [8,9], while France contributed to 1G with the Radiocom 2000 system [8,10]. Although they are predominantly based on analog technology, these 1G standards played a crucial role in the delivery of essential voice services. However, they faced limitations in terms of capacity, security, and efficiency [11,12].

The progression to digital technologies in subsequent generations, beginning with Second Generation (2G), led to significant improvements in communication quality and security and the introduction of data services in addition to voice communication [13]. Mobile communication networks entered the 2G era in the 1990s, revolutionizing telecommunications through widespread adoption of the Global System for Mobile Communications (GSM) standard. This advancement enabled digital encryption of data and voice signals, introduced features like roaming, and optimized spectrum utilization [14,15,16]. The 2G networks continued to evolve, resulting in so-called 2.5 Generation (2.5G) systems, which introduced a new packet switching technique and provided enhanced voice and data services to users [1]. The 2.5G networks brought important improvements that not only evolved the data rate but also improved established voice communications. These laid the foundations for high-performance mobile communications, laying solid ground for the next generations of mobile communication systems [17,18].

The constant evolution of wireless technologies led to an exponential necessity for high data rates and support for a large number of devices. Therefore, Third Generation (3G) mobile communication networks introduced mobile broadband services and improved data rates compared with previous generations [19,20]. In addition, 3G networks met the requirements of important vertical applications that require the exchange of high-definition images and videos, such as video calls and mobile Television (TV) [21,22]. The 4G mobile communication networks introduced a paradigm shift by adopting communication based on the Internet Protocol (IP). The 4G networks were meticulously designed to harness augmented bandwidths and elevated transmission data rates, allowing improved mobile broadband services, seamless web browsing, uninterrupted high-definition video streaming, substantially reduced latency, and a seamless handover [23,24,25,26]. These advancements were made possible through the development of cutting-edge technologies, such as Multiple-Input Multiple-Output (MIMO) and advanced signal processing techniques. Furthermore, 4G networks introduced OFDM in its physical layer, seamlessly aligning with the intricate demands for flexibility of 4G networks [1]. The use of the OFDM technique allows the system to use the available bandwidth efficiently and consequently makes the system less susceptible to interference, improving its spectral efficiency [27,28].

To meet the demands of an unprecedented number of connected devices, 5G mobile communication systems began to be implemented in 2020. These 5G networks will be able to support ultralow latency (∼1 ms), ultrahigh data rates (∼10 Gbps), high connectivity (∼1 million/km^2^) and reliability (∼10^−5^), and ultralow energy consumption. Furthermore, seamless integration of small cells and massive MIMO antennas has become essential to achieve the desired network capacity and coverage, especially in densely populated areas [29,30,31]. Additionally, 5G networks support communications at high frequencies (i.e., Millimeter Wave (mmWave) bands), which open up a range of possibilities, particularly in the context of the Internet of Things (IoT) [32]. Despite the ongoing deployment of 5G networks, both academia and industry have focused their attention on the development of 6G mobile communication networks, which are envisioned to present extraordinary advancements in terms of data rates, energy efficiency, latency, and reliability when compared with 5G networks. However, the architecture and performance components of 6G networks remain mostly undefined [33,34]. Despite this, to meet their anticipated requirements, existing communication systems will have to undergo a series of transformations that will allow them to operate under the conditions imposed by future 6G networks. These changes will affect all levels of communication, from the application layer to the physical layer. Within this transformative landscape, it is expected that the strategic integration of ML techniques plays a crucial role [35,36]. At the same time, a wide spectrum of modulation techniques are being investigated or proposed, including OFDM and its variants, to achieve a high level of spectral efficiency [37,38,39,40].

OFDM is the modulation technique for 4G and 5G wireless systems and is expected to remain indispensable to meet the rigorous demands of 6G networks [41,42,43]. Unlike their predecessors, 6G networks operate at higher frequencies and broader bandwidths, presenting distinctive challenges for the deployment of OFDM, particularly concerning issues such as a high PAPR and Inter-Carrier Interference (ICI), as outlined in [44]. In addition, OFDM is highly regarded for its ability to handle frequency selective fading, efficient spectrum usage, and resistance to interference [45]. However, it faces obstacles due to the close spacing of the subcarriers inherent in its design, which leads to a high PAPR. This elevated PAPR has several implications. First, it can result in intermodulation distortion, degrading the quality of the transmitted signal. Moreover, it decreases the efficiency of the amplifiers used in the system, as they must accommodate the high peaks by operating with a broader dynamic range. Finally, a high PAPR can cause unwanted signals to escape into neighboring frequency bands, causing interference with other wireless systems [46,47].

In scenarios where a high PAPR is not effectively managed, advanced signal processing and amplifier designs are needed, which can introduce additional delays in signal transmission and reception. These delays can lead to increased latency, which is undesirable in 6G networks that strive for ultralow-latency communication [48]. Consequently, optimizing the PAPR is crucial not only to mitigating interference, achieving high spectral efficiency, and reducing distortion, as previously mentioned, but also to obtaining low-latency signals in the context of 6G networks [49].

The literature is filled with a wide range of strategies designed to address the challenge of reducing the PAPR in communication systems [50]. These strategies encompass a spectrum of approaches, including conventional methods tested over time and more advanced techniques [51]. Some of these prominent techniques are clipping and filtering, Partial Transmit Sequence (PTS), Selected Mapping (SLM), interleaving, Tone Injection (TI), Tone Reservation (TR), and Active Constellation Extension (ACE). Clipping and filtering have gained recognition for their simplicity and effectiveness in mitigating the PAPR. However, they come with a notable caveat: the potential introduction of signal distortion, which can compromise the overall transmission quality [52,53]. PTS distinguishes itself through its ability to achieve substantial reductions in the PAPR. However, this method has a trade-off: it is a complex and computationally demanding approach that requires multiple iterations to achieve its objective [54,55]. SLM represents another powerful strategy for reducing the PAPR, with the added benefit of being less susceptible to signal distortion. However, it places significant demands on processing power and is sensitive to nonlinear channel conditions [56,57]. Interleaving exhibits effectiveness in PAPR reduction, particularly when used in conjunction with other techniques. However, its efficacy depends on specific configurations and may not yield optimal results when used alone [58,59]. Moreover, TI has emerged as an effective method for PAPR reduction, and it is less prone to distortion. However, to optimize its performance, it may require access to additional channel information [60]. TR shines in its high effectiveness in PAPR reduction, particularly in systems with nonlinear channels. However, it is a complex method that demands the allocation of additional tones, which can result in a reduction in spectral efficiency [61]. Finally, ACE stands out as another powerful tool in the arsenal for PAPR reduction, offering greater flexibility with different constellation schemes. However, it introduces some overhead and requires careful allocation of constellation points [62].

Motivated by the importance of the previously described PAPR techniques, in [50], the authors presented a review of the literature, recent research findings, simulations, and complexity analyses of these techniques. Organized into three primary categories for PAPR reduction schemes (i.e., signal distortion, multiple signaling, and coding), the paper presents a systematic approach to comprehending various methods that address the high PAPR challenge in OFDM systems. It serves as an invaluable resource for emerging researchers seeking a comprehensive understanding of the PAPR. However, the techniques discussed in [50] for PAPR reduction in wireless communication systems may fall short in addressing the complex and dynamic nature of the problem. Conventional approaches often rely on predefined signal processing strategies and parameter configurations which might not sufficiently adapt to the evolving conditions of wireless channels. While these methods may offer some degree of PAPR reduction, they might lack the adaptability required to consistently maintain optimal signal quality in the face of changing communication environments. Therefore, the authors of [50] acknowledged the significance of classical techniques, but they notably refrained from delving into ML approaches for PAPR reduction.

### 1.1. Contributions

To overcome the limitations of classical PAPR reduction techniques, recent works have revealed exciting prospects for applying ML to significantly reduce the PAPR while keeping the computational complexity at a minimum, which we will elaborate upon in more detail in Section 4, demonstrating that the use of ML has great potential for the field of wireless communications. Therefore, motivated by the importance of ML to 6G networks, this work presents a comprehensive review of PAPR reduction techniques based on ML. More specifically, this survey not only outlines various approaches but also presents the results, contributions, and limitations associated with several methods based on ML. In the process, this study endeavors to provide readers with comprehensive insights into the application of ML in this specific domain. In this context, it is worth highlighting that our research occupies a unique and indispensable position within the existing scholarly body of work because, to the best of our knowledge, this is the first work to present a review of ML-based techniques for PAPR reduction. Consequently, the main contributions of this paper can be summarized as follows:We present a comprehensive analysis of the classical PAPR solutions, highlighting their main features, advantages, and challenges;We present a concise review of the main PAPR reduction approaches based on ML and describe their main features, advantages, and challenges;We discuss the role of ML in the PAPR issue for 6G networks, giving directions for future work.

### 1.2. Research Methodology

To select the previously cited works, we considered the following methodology: (1) We identified relevant keywords that were aligned with the scope of the survey. These keywords included terms like PAPR, ML, and OFDM. (2) We performed comprehensive searches of several recognized academic databases, including Elsevier, IEEEXplore, and SpringerLink. We used the keywords defined in the first step to find potentially relevant articles. (3) The articles found were initially filtered based on titles and abstracts to ensure that they were directly related to our research scope. Articles that did not meet the criteria were excluded. Special care was taken to avoid articles published in scientific events without peer review or little recognition. (4) Finally, after the initial filtering, we proceeded to the full reading of the selected articles. This allowed us to make an assessment of their contributions and overall relevance. Some references from the selected works were separated for detailed reading and cataloging, feeding back into the process. To finish, Figure 1 illustrates the post-selection distribution of the selected articles across the databases.

To improve the readability of this survey, its organization is illustrated in Figure 2. The remainder of this paper is organized as follows. Section 2 presents a review of the main concepts of the OFDM modulation technique. Section 3 reviews the classical PAPR reduction techniques. Section 4 reviews the PAPR solutions based on ML. Section 5 describes the main challenges of applying ML techniques to PAPR optimization. Finally, Section 6 concludes the paper.

## 2. OFDM

Frequency Division Multiplexing (FDM) and OFDM are pivotal multiplexing and modulation techniques with deep historical roots and practical significance in contemporary wireless communication systems [63,64]. Their development over time has been driven by the imperative need to overcome a range of challenges associated with wireless data transmission, including issues such as interference, fading, and the constraint of limited bandwidth. These technologies serve as the foundation for numerous communication standards and are expected to continue to play a crucial role in the evolution of wireless communication systems [49,64,65].

More specifically, FDM is a classical multiplexing technique that splits the available frequency spectrum into multiple nonoverlapping bands, with each band allocated to a different channel or user. Each channel operates at a specific frequency, and multiple channels can coexist in the same communication medium without interference [66]. On the other hand, OFDM divides the available frequency spectrum into multiple orthogonal subcarriers, which are closely spaced, and each subcarrier is allocated a portion of the data, efficiently utilizing the spectrum. The orthogonality between subcarriers prevents interference, enhancing reliability [67]. Therefore, as illustrated in Figure 3, FDM modulation is less efficient in terms of spectrum utilization compared with OFDM [68].

In broadband communication systems, OFDM enables simultaneous data transmission by splitting the spectrum into multiple orthogonal subcarriers. The OFDM modulation relies on Inverse Fast Fourier Transform (IFFT) for transmission and Fast Fourier Transform (FFT) for reception. More specifically, the IFFT allocates data symbols to subcarriers, with a Cyclic Prefix (CP) added before transmission to avoid multipath effects, resulting in the signal x(n) [69]. The CP provides a guard period for the demodulation process, removing delayed echoes and facilitating accurate symbol detection in the frequency domain [70].

It is imperative to emphasize the importance of meticulously choosing the CP length with due regard to the expected channel delay spread [71]. Prudent selection of the CP length holds the utmost importance in wireless communication systems, especially in digital communication systems such as those used in mobile networks, Wi-Fi, and various wireless technologies [72]. This careful consideration plays a crucial role in minimizing the effects of temporal dispersion, ensuring reliable signal transmission, and enabling precise data reception. A well-considered choice for the CP length enhances the overall resilience and optimal functionality of the communication system [73,74]. For clarity, Figure 4 illustrates the OFDM transmitter/receiver system.

One of the main technical challenges inherent to OFDM systems is the emergence of an amplitude peak in the time domain due to constructive interference among multiple subcarriers. This intricate phenomenon is conventionally called the PAPR issue [75]. The PAPR has been a fundamental metric that characterizes the behavior of multicarrier signals, and it can be defined as [50]
(1)$$\mathrm{PAPR}=\frac{\max{\left|x(n)\right|}^{2}}{E{\left|x(n)\right|}^{2}}$$
where E|x(n)| is the mean value of the OFDM signal x(n) and max|x(n)| denotes the maximum amplitude peak of the OFDM signal.

As the number of carriers increases, phase accumulation arises among specific carriers during transmission amplification [76]. This results in power peaks (i.e., the amplitude of the signal reaches saturation levels which induce signal clipping and, consequently, result in abrupt discontinuities that add noise due to the superposition of multiple signals). In the context of OFDM, excessive noise can amplify interference between adjacent frequency bands, adversely affecting both the upper and lower bands. Therefore, a high PAPR value indicates a more pronounced deviation of the signal from its average power level, which results in lower amplifier performance and leads to height saturation and noise generation [77,78].

Figure 5 illustrates the Complementary Cumulative Distribution Function (CCDF) of the PAPR across different numbers of subcarriers (32, 128, 512, and 2048) with 16 Quadrature Amplitude Modulation (QAM) and with two of each subcarrier serving as pilot subcarriers, bearing the values −1 and 1. This arrangement resulted in a batch size of 1000, forming a matrix with 1000 rows and *N* columns. Notably, the pilot subcarriers were strategically placed in the second and *N*-2 columns. In addition, the CCDF is a valuable tool for assessing the reliability of a signal at various PAPR levels. This helps us understand the likelihood of the signal functioning correctly and maintaining its integrity.

Notably, from Figure 5, an observable trend emerges: higher subcarrier counts correlate with elevated PAPRs. This outcome is a consequence of the cumulative impact of subcarrier summations during the amplification process, as previously discussed. The graphical representation underscores the direct relationship between the number of subcarriers and the resulting PAPR, providing valuable insights into the system performance under varying configurations.

## 3. Classical PAPR Reduction Approaches

In the domain of OFDM systems, numerous techniques have been devised to mitigate the problem of the PAPR, which is a widespread issue in these systems. Such techniques can be classified into signal distortion approaches, signal scrambling methods, and hybrid and coding techniques [79,80]. However, this survey focused specifically on signal distortion approaches and signal scrambling methods. This is because our emphasis is on solutions that strike a harmonious balance between efficacy and simplicity, rendering them more accessible and applicable in practical implementations of OFDM systems [81]. Signal distortion approaches involve making direct modifications to the original signal in an effort to reduce the PAPR while striving to maintain the quality of the data [82]. These strategies include actions such as cutting, filtering, and ACE. In contrast, signal scrambling methods take an indirect approach to reduce the PAPR by carefully altering the signal without causing significant distortion to the data. Techniques falling within this category include SLM, PTS, interleaving methods, TI, and TR. The choice of technique to use depends on the specific requirements, trade-offs, and limitations of the particular communication system under consideration [80,83]. The main classical PAPR reduction techniques previously cited are described next.

### 3.1. PTS

The PTS technique is utilized in digital communication systems to address the challenge of a high PAPR in Transmit (Tx) signals without adding distortion to the signal. This method involves dividing the input data stream into multiple sub-blocks which are independently modulated. After splitting the data stream, the Inverse Discrete Fourier Transform (IDFT) of each sub-block is obtained and weighted by a phase factor which is optimized to minimize the PAPR of the sum of the signals of all sub-blocks [84]. The main step of the PTS technique is the phase optimization process, which strategically manipulates the phases of the sub-blocks within the signal without altering the original Tx signal [85]. In this technique, the reduction in the PAPR is dependent on the sub-blocks’ size and the number of allowed phase factors. Despite its effectiveness in mitigating PAPR issues, the PTS computational complexity can pose challenges, particularly depending on the modulation order [86,87,88]. In addition, it is important to highlight that the PTS technique needs some bits of side information to be transmitted, conveying the phase factors, which reduces its spectral efficiency.

Integration of the PTS scheme into OFDM systems is pivotal for reducing the PAPR and enhancing Bit Error Rate (BER) performance. Figure 6 visually depicts this integration, illustrating how the PTS scheme is implemented within an OFDM system [89,90]. In Figure 6, the phase optimization is particularly emphasized after the IDFT operation [91]. By judiciously selecting these phase coefficients, the PTS technique minimizes constructive interference, leading to an overall reduction in the PAPR of the Tx signal. This reduction is highly desirable as it not only enhances BER performance but also improves the reliability and resilience of the wireless communication system [87,88].

### 3.2. SLM

In SLM, the process of generating alternative signals from a single information signal involves a systematic and adaptive approach. This involves creating a set of potential signals, called “candidate signals”, by introducing variations based on different phase sequences. These candidate signals are then subject to adaptive optimization techniques, and then the signal with the minimum PAPR is transmitted [92]. To elaborate further, think of these candidate signals as alternative versions of the original signal, with each slightly modified in a systematic way. The variations are introduced in a controlled manner to cover a range of possibilities. Then, adaptive optimization techniques come into play to dynamically refine these candidate signals during the transmission process [93,94].

Once generated, the signals undergo a thorough evaluation with careful scrutiny to identify the one with the lowest PAPRs. This comprehensive method ensures both the efficiency and reliability of the transmission. The ultimate goal of SLM is to minimize signal distortion and enhance power efficiency, a critical aspect of the overall performance of the communication system. Furthermore, the selected phase sequence needs to be transmitted as side information so that the receiver can recover the original sequence, leading to a reduction in the spectrum efficiency. Moreover, it is important to note that the computational complexity of SLM increases with the number of candidate signals generated [94,95]. Each signal requires meticulous evaluation and comparison, increasing the computational overhead, especially when dealing with a large number of candidates [96].

In essence, SLM involves a careful and adaptive exploration of potential signal variations to optimize the transmission process, ensuring both efficiency and reliability in communication systems. Figure 7 provides a visual representation of this approach in the Tx, highlighting the strategic incorporation of signals before the IDFT process to optimize power efficiency within the signal. These preemptive signals are meticulously crafted to alleviate the elevated PAPR encountered in the transmitted signal [94,97,98]. Moreover, from Figure 6 and Figure 7, it is possible to verify that the phase variation is applied at different points in the transmitter in PTS and SLM, producing different results but with the same goal.

SLM’s effectiveness is attributed to its capability to substantially decrease the PAPR while causing a smaller impact on the spectral efficiency with respect to PTS. The direct approach of SLM and its proven success in PAPR reduction position it as a preferred method for striking a balance between reducing the PAPR and preserving spectral efficiency in OFDM communication systems [94,97,98].

### 3.3. Interleaving

This PAPR reduction technique is similar to the SLM and PTS approaches, but in the interleaving technique, interleaves are considered to generate multiple signals instead of phase sequences. More specifically, the interleaving technique rearranges the order of the transmitted symbols to decrease the chances of encountering high-power peaks. Then, the signal with the smallest PAPR is chosen for transmission. By interleaving the symbols, the high-power peaks are more evenly spread out, reducing the likelihood of their occurrence [99]. This enhances the BER performance by mitigating the impact of significant power variations. Moreover, interleaving does not introduce additional signal distortions or require intricate processing, ensuring a low BER [100]. However, to correctly decode the transmitted signal, it is necessary to transmit some side information bits to inform the receiver which interleaver was considered to generate the selected signal for transmission [99,100].

Regarding complexity, the interleaving process is generally straightforward, typically involving symbol permutation according to a predefined pattern. Its computational complexity is usually low, making it a practical choice for implementation in communication systems [101]. However, it is essential to note that the interleaving technique cannot achieve the same level of PAPR reduction as more advanced techniques like SLM. Therefore, the achieved PAPR reduction through interleaving should be carefully balanced with the desired BER performance. In comparison with SLM and PTS, interleaving introduces a distinct dimension to signal processing, emphasizing error resilience and the ability to recover from channel-induced disruptions [99,102]. Figure 8 illustrates a block diagram of the OFDM transmitter with the interleaving approach. This visual representation provides a clear overview of how the interleaving technique is applied in an OFDM system to reduce the PAPR and improve BER performance [99].

### 3.4. Clipping and Filtering

Clipping and filtering play a pivotal role in the management of both signal characteristics and system performance. This PAPR reduction approach limits the signal power to a predetermined Clipping Level (CL) by using a clipper (i.e., if the signal exceeds the CL, then the signal is clipped; otherwise, the signal does not suffer any change). Clipping serves as a highly effective approach to governing signal amplitudes and diminishing the PAPR without the need for side information at the receiver [103]. However, although the clipping strategy is performed at the transmitter, the receiver must estimate the location and size of the clip, which is difficult in practice and results in the introduction of nonlinear distortions, thereby bearing implications for BER performance. The distortions that arise from clipping can lead to an increased BER, particularly when the clipping threshold is aggressively configured [103,104].

In contrast, filtering the signal after clipping is a key solution to reducing these distortions and consequently increasing the overall signal quality [105]. Filtering can eliminate out-of-band distortions, thereby fostering spectral efficiency and curtailing interference with adjacent channels [106]. Nonetheless, it is crucial to strike a balance between mitigating out-of-band distortions and reducing the peak power of the signal, as excessive filtering may give rise to power inefficiencies [107]. Furthermore, it is worth noting that the intricacy of the clipping and filtering processes can impose a high computational demand on the system due to the incorporation of advanced filtering algorithms or iterative procedures. Consequently, system designers must evaluate the trade-offs between computational complexity, BER, and power efficiency during the implementation of these approaches [108,109].

The above considerations underscore the importance of carefully configuring and optimizing the clipping and filtering processes to attain the desired signal quality while reducing the system complexity, BER, and power consumption [106]. Figure 9 depicts the implementation of clipping and filtering at the OFDM transmitter to manage the PAPR, optimize BER performance, and enhance signal quality [110].

### 3.5. TI and TR

TI is an approach that involves the injection of specific tones or subcarriers into the transmitted signal to mitigate high-power peaks [111]. The primary goal of TI is to strategically redistribute the power peaks across the signal, resulting in a reduced PAPR [112]. One of the key advantages of the TI method is the ability to improve the BER performance while maintaining relatively low complexity. In addition, the TI technique does not need side information at the receiver. Therefore, the TI approach introduces minimal signal distortions and can be implemented with modest computational resources [113,114]. On the other hand, TR is a distinct technique that focuses on allocating a portion of the available subcarriers in an OFDM system for the inclusion of additional tones. These reserved tones are carefully designed to nullify high-power peaks in the transmitted signal. The TR method aims to enhance the BER performance by mitigating the effects of significant power variations [115,116]. Similar to the TI approach, TR can be implemented with relatively low complexity, as it involves generating and adding reserved tones to the signal [117]. However, different from the TI method, the TR approach requires additional side information at the receiver to correctly decode the received signal [117].

In addition, TI is known for its efficient redistribution of power peaks, which directly contributes to reducing the PAPR and improving the BER. Moreover, TI introduces minimal distortions and is suitable for practical applications [118]. On the other hand, TR offers a dedicated approach by reserving specific subcarriers for additional tones, further reducing the impact of power variations on the transmitted signal. The choice between TI and TR should be made based on the specific requirements and characteristics of the communication system [119,120]. TI and TR are not one-size-fits-all solutions, and their effectiveness can vary depending on factors such as the number of reserved tones, their positioning, and the prevailing channel conditions [121]. Both techniques require careful consideration of the trade-off between PAPR reduction and BER performance [122]. Finally, Figure 10 and Figure 11 illustrate the block diagram of the OFDM transmitter for TR and TI, respectively [123]. From these figures, it is possible to verify that, different from TI, the TR approach adds nulls in the data subcarrier positions to avoid the high-power peaks in the transmitted signal [50]. The presence of zeros in TR serves to generate interference patterns, aiding in a reduction in the PAPR when integrated with the primary data signal. This forms a crucial component of the design approach aimed at efficiently mitigating or minimizing peaks within the transmitted signal. In contrast, TI accomplishes PAPR reduction by introducing artificial tones directly into the primary data signal. The lack of zeros between data subcarriers streamlines the injection of these tones, and optimization algorithms ensure that these artificial tones interact seamlessly with the data subcarriers [50,120,122,123].

### 3.6. ACE

The ACE technique is used to expand the constellation points within a transmitted signal to reduce the impact of high-power peaks [124]. More specifically, the constellation points are extended toward the outside of the original constellation, and regions named “feasible regions” are created as illustrated in Figure 12 [123]. This expansion of constellation points enlarges the signal space, resulting in a more even distribution of signal amplitudes and, consequently, a lower PAPR and BER due to reduction of the influence of significant power fluctuations [125]. Furthermore, the ACE technique does not require any side information at the receiver and does not add distortions in the signal as it is considered a signal scrambling approach. Implementation of the ACE technique presents moderate complexity and requires adjustments to the way constellation points are mapped and processed [126,127]. ACE introduces some additional computational load, as expanding constellation points requires careful design considerations and may require iterative optimization algorithms [128]. Nevertheless, its complexity remains manageable and practical for implementation [129].

It is crucial to understand that the effectiveness of ACE in reducing the PAPR, and its impact on the BER depends on various factors [130]. These factors include the specific constellation design, Signal-to-Noise Ratio (SNR), and channel impairments. When selecting and optimizing extended constellation points, a balance must be struck between reducing the PAPR and maintaining BER performance [131,132]. In some situations, expanding constellation points can introduce additional signal distortions, which could potentially lead to an increased BER. Therefore, a trade-off between PAPR reduction and the BER needs to be carefully considered [133].

We finish with Table 1, which summarizes the main features of the aforementioned PAPR reduction techniques. More specifically, Table 1 compares the classical PAPR techniques in terms of computational complexity, BER escalation, the need for side information, and whether the technique adds distortion to the signal or not. Evaluating these factors is crucial for determining the efficacy and cost-benefit of a given method [132]. In addition, for more details about the classical PAPR reduction techniques, readers should consult [50].

## 4. Role of ML in PAPR Reduction

The techniques discussed in Section 3, covering PAPR reduction in wireless communication systems, may fall short in addressing the complex and dynamic nature of the problem. Conventional approaches often rely on predefined signal processing strategies and parameter configurations that might not sufficiently adapt to the evolving conditions of wireless channels. While these methods may offer some degree of PAPR reduction, they might lack the adaptability required to consistently maintain optimal signal quality in the face of changing communication environments. On the contrary, ML plays a key role in overcoming these limitations. It offers a more sophisticated and adaptive approach to PAPR reduction in OFDM systems [134]. ML algorithms demonstrate remarkable capabilities in finding intricate patterns within PAPR levels, making accurate predictions about future PAPR levels [135,136]. The ability to dynamically adjust transmission parameters in real time further distinguishes ML techniques, ensuring a continuous optimization process that minimizes PAPR levels while preserving the optimal signal quality.

Moreover, ML enables exploration in a vast solution space of signal processing strategies and parameter configurations, leading to the identification of high-performance approaches for PAPR reduction [137]. Through algorithmic optimization and feature selection, ML assists in the development of PAPR reduction strategies that strike a balance between performance and complexity. This is crucial for practical implementation in real-world communication systems. The adaptability of ML techniques ensures seamless integration of PAPR reduction strategies within dynamic communication systems [138]. This adaptability adapts to the evolving requirements of communication systems, accommodating changes in the environment and communication protocols [139]. Therefore, ML not only assists with reducing the PAPR but also contributes to achieving notable enhancements in signal quality by minimizing signal distortion and optimizing power consumption [140].

### 4.1. ML Technique Review

Motivated by the above cited advantages, several works in the literature explored different ML techniques [134,135,141,142,143,144,145,146,147,148,149,150,151,152,153,154,155] to reduce the PAPR. Therefore, to improve the understanding of readers not particularly familiar with ML, we next present a brief explanation of the ML techniques considered by the surveyed papers.

#### 4.1.1. Artificial Neural Network (ANN)

An ANN is an ML technique that is inspired by the functioning of the human brain. It is made up of fundamental units called neurons, which are organized in layers. The input layer receives the initial data and passes them to the next layer. Each neuron in this layer represents a specific feature of the input data. Hidden layers are intermediate layers between the input layer and the output layer. Each neuron in a hidden layer processes information from previous neurons and forwards it to the next one. The output layer generates the final result of the neural network. This output can be a rating, a continuous value, or any other form of representation, depending on the task performed by the network. Each connection between neurons has an associated weight, determining the strength and direction of the influence of one neuron on the other. During training, the network adjusts these weights to minimize the difference between the predicted outputs and desired outputs using a process called backpropagation. This involves calculating the error at the network output and adjusting the weights back across layers to improve prediction accuracy. This process is repeated iteratively with training datasets until the network achieves satisfactory performance [156,157].

#### 4.1.2. Deep Neural Network (DNN)

A DNN with encoder and decoder architecture is often used in Deep Learning (DL) tasks, such as autoencoder models, and also in more complex architectures, such as those used in convolutional neural networks. More specifically, the encoder presents five components: (1) an Fully Connected (FC) layer, which is a dense layer that connects each neuron to all neurons in the next layer, making it easier to capture global patterns in the data; (2) Batch Normalization (BatchNorm), a component which normalizes the activation values, reducing internal covariance and consequently accelerating training and improving generalization; (3) Rectified Linear Unit (ReLU), an activation function that introduces nonlinearity to the model, allowing the incorporation of complexity between layers; (4) another FC layer, namely a dense layer which possibly has fewer neurons than the previous layer, helping to compress information; and (5) a sigmoid function, which is a final activation function in the encoder layer. It can be used to generate values between 0 and 1, being useful in binary classification problems or representing probabilities [158].

In addition, the decoder is composed of the same components as the encoder, but in the decoder, the FC layer receives the compressed representation of the encoder data and seeks to reconstruct the original data. Similar to the BatchNorm layer in the encoder, it normalizes the activation values during training to speed up convergence. ReLU adds nonlinearity to the network, enabling the learning of complex relationships in the data. The FC layer generates the final output of the decoder, which must be a reconstruction of the original data. BatchNorm normalizes the activation values before the last ReLU layer in the decoder. ReLU is the last activation layer to introduce nonlinearity in the decoder output [159].

#### 4.1.3. Monte Carlo Tree Search (MCTS)

The MCTS emerges as an innovative approach to deal with complex search problems in vast and little-explored decision spaces. This method is based on stochastic simulation and adaptive evaluation, offering an effective solution for situations where an exhaustive search of all possibilities is impractical. In simple terms, MCTS represents the problem as a tree, where each node represents a possible state and the edges indicate achievable actions. Starting from a root node that represents the initial state of the problem, MCTS follows an iterative process that involves selection, expansion, simulation, evaluation, and backpropagation. This adaptive approach allows one to explore the decision tree efficiently, focusing efforts on the most promising areas of the search space. MCTS excels in scenarios in which the absence of an explicit model of the environment or an extensive search space makes exhaustive searching impractical. By incorporating stochastic simulations and dynamic evaluations, MCTS proves to be a useful tool for solving complex problems, providing practical and adaptable solutions in diverse applications [160].

#### 4.1.4. Deep Unfolding Network (DUN)

A DUN is a specific type of model-driven Neural Network (NN) that operates in layers. Each layer performs progressive analysis of the data, learning more complex representations over time. It is interesting to think of this as a series of observation steps, starting with simple details and gradually moving to more abstract features. This process allows the network to understand patterns and become effective in performing complex tasks [161].

#### 4.1.5. Random Sample Consensus (RANSAC)

RANSAC is an iterative ML method that is used to estimate the parameters of a mathematical model considering a database that contains outliers. This non-deterministic method is widely used when there is a large amount of noisy data, and the main goal is to design a model that fits the data well. Then, RANSAC can be used to classify the remaining data as either inliers or outliers until a satisfactory model is found [134].

Finally, the next subsection presents a concise review of PAPR reduction solutions based on ML as documented in the literature. This exploration aims to provide insights into the diverse range of ML techniques employed to address the challenges associated with the PAPR in wireless communication systems.

### 4.2. PAPR Reduction Approaches Based on ML

As previously mentioned, ML is a key to solving the PAPR issue in OFDM systems. Therefore, several works in the literature have studied the application of different ML techniques to reduce the PAPR [134,135,141,142,143,144,145,146,147,148,149,150,151,152,153,154,155]. Therefore, as the main goal of this paper is to show the role of ML in PAPR reduction, we next present the reviewed papers, which are organized following the considered ML area, and describe their main contributions, features, and limitations.

#### 4.2.1. ANN [135,141,142,143,144,145,146]

The main ML technique considered in the literature to solve the PAPR reduction problem is the ANN [135,141,142,143,144,145,146]. These proposed solutions present the same main steps, which are illustrated in Figure 13. More specifically, when an ANN technique is considered, the OFDM signal generated by the IFFT is the input of the ANN, which is composed of multiple FC layers. For each layer, a weight matrix and a bias matrix are applied to find the weighted sum of the inputs, and then an activation function is considered. In a supervised training process, these weights are optimized with the goal of minimizing a cost function, as with the PAPR. Then, in regular operation, the trained ANN is used to generate an OFDM signal with a low PAPR.

After understanding the basic operation of an ANN-based PAPR reduction solution, we can discuss in more detail the main approaches in the literature [135,141,142,143,144,145,146]. More specifically, in [141], an innovative approach is proposed to mitigate the PAPR in a Visible Light Communication (VLC) Direct Current-biased Optical OFDM (DCO-OFDM) system, employing a deep autoencoder. This method leverages a deep autoencoder to diminish the PAPR by inputting the constellation symbols into the autoencoder. When introducing a novel PAPR reduction scheme for VLC DCO-OFDM, the performance of this approach was scrutinized in comparison to conventional methods using CCDF. The proposed deep autoencoder demonstrated a notable 6.5 dB reduction in the PAPR within the VLC DCO-OFDM system while preserving an acceptable BER and efficiently curtailing the PAPR for high-speed data transmission. In [142], the authors introduced a novel method called Neural Network ACE (NN-ACE), and this paper aims to mitigate the PAPR in OFDM systems. NN-ACE involves extending the constellation using an autoencoder, where the autoencoder is trained to expand the input symbols in the complex domain, resulting in a reduced PAPR. In diverse channel conditions, NN-ACE demonstrates similar effects in extending Quadrature Phase Shift Keying (QPSK) constellations and reducing the PAPR compared with traditional ACE methods. However, it surpasses conventional schemes. When analyzing the CCDF curves of the PAPR and the symbol power, it is possible to note that while the ACE method decreases the PAPR, it comes at the expense of increased signal power. On the other hand, NN-ACE achieves a lower PAPR but at the cost of higher power consumption. Notably, when NN-ACE is implemented, there is a degradation in the overall BER performance.

In addition, the authors of [143] presented an innovative approach to improving the efficiency of indoor VLC-OFDM systems. It addresses challenges like a high PAPR and Light Emitting Diode (LED) nonlinearity by using a unique PAPR reduction scheme involving weighted autoencoder and amplitude clipping techniques. DL is applied for adaptive acquisition and optimization of the constellation mapping, de-mapping, and phase factors. The study systematically investigated the impact of hyperparameters, network architecture, and channel types on performance metrics, focusing on the BER and PAPR. The results show that the hybrid autoencoder method achieved a significant PAPR reduction of about 12 dB, demonstrating resilience to LED nonlinearities and improved BER performance compared with conventional methods. Tailored for Layered Asymmetrically Clipped OFDM (LACO-OFDM)-based VLC systems, the hybrid autoencoder network adapts to diverse channel conditions, leading to a substantial reduction in the PAPR. The simulation results suggest seamless integration into typical VLC-OFDM systems, offering favorable PAPR and BER performance even in the presence of Inter-Symbol Interference (ISI) in the diffused optical wireless channel. While acknowledging limitations in real-world implementations and variations in system conditions not fully captured in simulations, the study provides valuable insights into addressing challenges in indoor VLC-OFDM systems.

In [144], the mitigation of the PAPR in a Generalized Frequency Division Multiplexing (GFDM) system was accomplished through the utilization of an encoder-decoder NN, commonly referred to as an autoencoder. The assessment of the proposed PAPR-Reducing Network (PRnet) strategy was carried out by analyzing the CCDF of the PAPR. The effectiveness of this approach was substantiated through a comparative analysis against conventional methods, illustrating its superior capability in PAPR reduction. Moreover, the authors of [145] introduced a new method for mitigating the high PAPR issue in wireless transmitters using OFDM systems. The proposed method utilizes a real-valued NN, which consists of a PAPR reduction module and a PAPR decompression module. These modules are jointly trained offline to simultaneously reduce the PAPR and minimize the BER. At the receiver, the PAPR decompression module aids in signal reconstruction to minimize the BER without introducing significant distortion. Through extensive simulations, the method demonstrated superior performance in terms of the PAPR and BER compared with existing methods like clipping, the NN model, and PRnet. Notably, there was a reduction in the BER of 3.5 times at a 20 dB SNR. Importantly, the proposed method exhibited lower computational complexity than PRnet, making it a promising solution. It is crucial to acknowledge the limitations and specify scenarios where the proposed method remains applicable, such as in MIMO systems, without impacting the overall system structure.

The work in [135] aimed to tackle the challenge of nonlinear relationships between the intended and actual PAPR in frequency-selective PAPR reduction. The proposed solution, PAPRer, employs a novel ML-based approach to automatically and precisely adjust the optimal PAPR target. This involves utilizing features related to the clipping noise filter and minimizing a defined loss function through supervised learning. In the context of 5G New Radio, PAPRer was demonstrated through numerical evaluations to predict and adjust optimal PAPR targets with high accuracy. The complexity assessment presented indicates that PAPRer strikes a favorable balance between performance and complexity. Moreover, the article proposes an ML-based method specifically addressing Inter-Carrier Waveform (ICWEF). It highlights the capability of this method to accurately predict optimal PAPR targets. Moreover, the authors of [146] emphasized the LSTM-Autoencoder (LSTM-AE) model for its ability to handle variable input sequential data and its flexibility in finding a compromise between the PAPR and BER through hyperparameter tuning. This paper aims to tackle the PAPR issue in OFDM systems specifically within VLC. The proposed solution involves employing an LSTM-AE, a combination of an autoencoder and Long Short Term Memory (LSTM), to efficiently learn a concise representation of the input and accommodate variable-length sequences. The proposed LSTM-AE model presented superior performance in terms of the PAPR compared with various existing strategies, all while maintaining a balance with the BER. The conclusion asserts the model’s superiority in simulations over traditional schemes. Nevertheless, the text acknowledges certain limitations, underscoring the importance of further research, particularly in the development of faster training methods for larger OFDM systems with increased subcarriers and modulation orders.

#### 4.2.2. DL [147,148,149,150,151,152,153,154]

Another area of ML largely considered to solve the PAPR optimization problem is DL [147,148,149,150,151,152,153,154]. As for the ANN-based approaches, the DL-based solutions usually consider the same basic operation [147,148,149,150,151,152,153,154]. Therefore, the basic blocks of the main DL technique considered in the literature (i.e., DNN) is illustrated in Figure 14, where it is possible to verify that, different from the ANN, a DNN based on an autoencoder (encoder/decoder) is composed of the cascaded sublayer which contains FC dense layers, normalization algorithms, and activation functions. These sublayers are responsible for learning, through a supervised training process, an efficient abstract representation of the OFDM signal generated by the IFFT and generating a new OFDM signal based on superimposing the output signals. The generated OFDM signal should present a low PAPR.

Among some relevant works considering DL techniques to reduce the PAPR in OFDM systems [147,148,149,150,151,152,153,154]. In [147], the authors employed a DL paradigm featuring an autoencoder network that proved to be highly effective in addressing the nonlinearity induced by LED in VLC. The transmitter integrates a DNN employing Discrete Fourier Transform Spread (DFT-S), while the receiver is bifurcated into two subnets. The cost function, encompassing both autocorrelation and the mean square error, contributes to the scheme’s superior performance in terms of the BER and training speed. This model-driven DL approach incorporates an autoencoder network with distinct encoder and decoder subnets for symbol mapping and detection. To tackle LED nonlinearity within VLC employing OFDM, an autoencoder-based NN, enriched with expert domain knowledge, was deployed. The architecture and loss function of the autoencoder network were meticulously designed to encapsulate expert knowledge, and an investigation into the impact of the regularization parameter and training SNR on the network was conducted. The proposed NN, infused with expert domain knowledge, adeptly mitigated LED nonlinearity, nonlinear distortions, and ISI while remaining compatible with CP removal. It efficiently addressed interference stemming from multipath channels, thereby enhancing overall performance. Moreover, in [148], the innovative Tone Reservation Network (TRNet) scheme, grounded in DL principles, was devised to effectively diminish the PAPR. TRNet exhibits adaptability to input signals, strategically generating signals that neutralize peaks. Through simulations, it was evident that the proposed TRNet outperformed the other strategies in terms of PAPR reduction, achieving notable results with a reduced number of reserved tones. The efficacy of TRNet in PAPR reduction within OFDM signals was convincingly demonstrated. Notably, TRNet contributed to accelerated training speeds while maintaining commendable BER compensation. This DL-based TRNet scheme showcased its superiority in PAPR reduction with fewer tones, ultimately enhancing bandwidth efficiency in OFDM systems. The utilization of an Feedforward Neural Network (FFNN) within TRNet further solidified its capability to generate peak-canceling signals, further emphasizing its effectiveness in addressing PAPR challenges.

A novel DL algorithm guided by a model was introduced in [149] to effectively mitigate the PAPR in OFDM systems. This algorithm incorporates an iterative peak-canceling signal generation scheme within a DL network. The trainable parameters within this scheme optimize both the clipping threshold and the weights of the time domain kernel function, resulting in comparable PAPR performance to existing approaches. Notably, this algorithm achieves these outcomes with a commendable reduction in complexity and training costs. Furthermore, the proposed tone reservation algorithm, grounded in a model-driven DL approach, is specifically designed for PAPR reduction. The fusion of a physical model with DL enhances its overall performance and provides a more insightful interpretation of the results. This innovative approach represents a significant stride in addressing PAPR challenges, combining efficiency with a more comprehensive understanding of the underlying processes. Moreover, a groundbreaking adaptive modulation scheme attuned to the intricacies of the PAPR was innovatively introduced in [150] to enhance the energy efficiency of OFDM. The task of maximizing energy efficiency was aptly framed with meticulous considerations for power and modulation constraints. Confronted with the non-convex nature of this optimization challenge, a cutting-edge solution grounded in online DL was proposed. This pioneering protocol demonstrated a substantial increase in energy efficiency, achieving a remarkable gain of up to 3 dB. Notably, in comparison with conventional PAPR-unaware protocols, the PAPR-aware adaptive modulation scheme reduced the PAPR by an additional 3 dB. The methodology hinges on the application of online DL for optimization purposes.

In [151], the authors delved into an in-depth exploration of a DNN-based receiver tailored to LACO-OFDM systems. In this comprehensive analysis, various schemes were meticulously compared, scrutinizing their spectral efficiency and complexity. The study also addressed the determination of the minimum number of FC layers essential for optimal signal detection. Introducing a novel approach termed O-OFDMNet, the paper leveraged DL techniques for encoding and detection in LACO-OFDM, DCO-OFDM, and low-complexity LACO-OFDM systems. Impressively, O-OFDMNet achieved throughput levels akin to Radio Frequency (RF)-OFDM, maintaining a moderate level of complexity. Noteworthy is its superior resilience to practical optical channel conditions when benchmarked against existing counterparts. The DL-enhanced O-OFDMNet not only matches the spectral efficiency of classical RF-OFDM but also stands out as a power-efficient solution. It succeeds in reducing the BER and PAPR, concurrently enhancing the capacity. Furthermore, O-OFDMNet exhibits robustness against noise and distortion, showcasing a shortened runtime for added efficiency. Moreover, the authors of [152] proposed a DUN called the PR-DUN model, which was intricately crafted to address the challenge of the PAPR in OFDM systems, employing a technique known as deep unfolding. This model dynamically adjusts its parameters through backpropagation and minimizes a loss function intricately linked to the PAPR. Notably, the PR-DUN model accommodates any transmit power constraint, providing fine-tuned control over power increases. These distinctive features collectively empower the PR-DUN model to achieve a more substantial reduction in the PAPR alongside a diminished BER. It is worth highlighting the commendable computational efficiency of the PR-DUN model when compared with other similar solutions. The incorporation of trainable parameters and a stochastic optimization algorithm allows the model to optimize its variables for effective PAPR reduction. The utilization of the ReLU activation function within its layers adds to the model’s adaptability and performance.

A method leveraging soft clipping was introduced in [153] for Tensor-train Residual Deep Neural Network (TT-RDNN), aiming to deterministically diminish the PAPR. The traditional Residual Deep Neural Network (RDNN) is replaced with a TT-RDNN, resulting in a reduction in the parameters and memory usage. The efficacy of the TT-RDNN was thoroughly assessed across diverse modulation alphabets. A pioneering DL architecture was proposed to strategically lower the PAPR of OFDM systems. This innovative architecture ensures a linear power amplifier response with a deterministic threshold while minimizing performance loss. The incorporation of tensor-train decomposition in the architecture not only reduces the parameters but also accelerates training and lowers memory usage. The outcome was a DL model that significantly lowered the PAPR of OFDM, adhering to spectral mask requirements and minimizing performance loss. In summation, a learning-driven strategy was harnessed to jointly formulate the transmit and receive filters in communication systems [154], with the primary goal of maximizing the achievable information rates while adeptly managing the Adjacent Channel Leakage Ratio (ACLR) and PAPR. In stark contrast to conventional techniques, this approach achieved a substantial reduction in both the ACLR and PAPR, paving the way for the transmission of competitive rates within a 3rd Generation Partnership Project (3GPP) multipath channel. Crucially, this learning-based method seamlessly integrates into the transmitter without introducing any supplementary complexity, rendering it particularly advantageous for 6G networks. The joint design of transmit and receive filters is facilitated by an NN-based detector for waveform learning. The method not only optimizes waveform design for communication systems, ensuring competitive rates with diminished ACLRs and PAPRs, but it also enables the multiplexing of multiple users through joint optimization. Importantly, it achieves these enhancements without a significant loss in the information rate when compared with conventional baselines. Furthermore, a reduction in the ACLR and PAPR is achieved without adding complexity to the transmitter, underscoring the efficiency and practicality of this proposed approach.

#### 4.2.3. Other ML Techniques [134,155]

In addition to previously considered ML techniques, a few works in the literature explored different ML techniques. For example, in [155], the authors presented a low-complexity method for reducing the PAPR in OFDM signals using MCTS. The MCTS strategy is used to create a tree structure that represents the searching space and searches for the optimal solution by sequentially finding a path from the root node to a leaf node. Figure 15 illustrates the MCTS strategy for PAPR reduction, from which it is possible to observe that the MCTS receives the OFDM signal generated by the IFFT or another classical PAPR reduction method, and after this, the MCTS is used to search for the optimal phase vector and generate a new OFDM signal with a low PAPR. More specifically, the authors outlined proposals for a low-complexity approach aimed at mitigating the PAPR of OFDM signals. The proposed method leverages the MCTS to enhance PAPR reduction, emphasizing a delicate equilibrium between minimizing the PAPR and efficiently managing computational load. In addition, to introducing the proposed low-complexity PTS method, the authors also considered the conventional reduced complexity PTS method and the iterative flipping PTS method. They further introduced the Monte Carlo PTS (M-PTS) method, which integrates the MCTS to reduce the PAPR in OFDM. In addition, this paper underscores the novel combination of the MCTS with dominant PTS methods as an effective strategy for complexity reduction. According to the proposal, this method significantly decreases computational complexity by a factor of 35, claiming a noteworthy 0.2 dB enhancement in the PAPR at half the computation cost. However, it is crucial to note that the proposed method does exhibit a PAPR performance degradation of 1.16 dB.

Finally, the authors of [134] introduced a novel ML technique for the purpose of diminishing the PAPR in massive MIMO OFDM systems. This approach optimizes the PAPR by considering a model fitting method named RANSAC. This optimization process is conducted by generating multiple candidate approximations obtained by randomly subsampling the grid points. Then, RANSAC is considered to detect the number of outliers (larger distance of the considered optimization function) in each generated candidate. Therefore, the determined best model is the one that minimizes the number of outliers. In addition, PAPR reduction is also accomplished through diverse methods, including clipping and filtering, ACE, TR, Selective Tone Reservation (STR), and Unused Beam Reservation (UBR), as summarized in Figure 16. Additionally, the DNN and Self-Organized Map (SOM) techniques are integrated into the process. This ML-based approach proves effective in significantly reducing the PAPR within a massive MIMO system employing OFDM. To alleviate the laborious task of manually tuning the hyperparameters for various scenarios, the introduction of an optimal hyperparameter function computation emerges as a practical solution, streamlining the optimization process.

Finally, from the previous detailed discussion about the ML-based PAPR reduction approaches, we can say that the spectral efficiency, computational complexity, BER, and PAPR are crucial elements, each presenting specific challenges. Finding the ideal ML approach demands a balance between these factors while considering their interactions. Each technique has different characteristics and performance levels in several aspects. Therefore, choosing the most appropriate technique depends on the specifics of the evaluated scenario and its requirements. In some situations, the complexity of the process increases, but the performance in terms of PAPR reduction stands out, ultimately paying off. On the other hand, in different contexts, an increase in complexity cannot be tolerated, or even some latency constraints influence the decision.

The applicability of each technique depends on the specific needs of each implementation. It is essential to consider the trade-offs between the spectral efficiency, computational complexity, BER, and PAPR, seeking to find the ideal balance to meet the objectives of the communication system in question. Moreover, we can also verify that each work addresses a distinct system model, incorporating specific modulation schemes and varied OFDM parameters. Due to these differences, a direct comparison between the amount of PAPR reduction that each one attains would be unfair and even potentially misleading, as each approach has its own particularities, including the considered level of BER degradation.

Therefore, Table 2 summarizes the performance of all reviewed ML-based methods in terms of PAPR reduction when compared with the classical approaches presented in Section 3. From Table 2, it is possible to verify the superiority of the ML-based approaches when compared with classical PAPR methods, showing the importance of studying new ML techniques to solve the emerging PAPR reduction problem. To finish, this table serves as a valuable resource for researchers and practitioners looking to explore and compare different ML-based PAPR reduction methods. However, each technique has different characteristics and performance levels in several aspects. Therefore, choosing the most appropriate technique depends on the specifics of the evaluated scenario and requirements. In some situations, the complexity of the process increases, but the performance in terms of PAPR reduction stands out, ultimately paying off. On the other hand, in different contexts, an increase in complexity cannot be tolerated, or even some latency constraints influence the decision. Therefore, it is quite hard (if possible) to answer which one is preferable. It is crucial to carefully evaluate the individual needs of each situation before deciding which technique to adopt. Finally, Table 3 summarizes in chronological order the PAPR solutions based on ML cited previously, providing a comprehensive understanding of their proposed methods and goals. In addition, Table 4 describes the main contributions and limitations of the previously described works.

### 4.3. Limitations of ML-Based PAPR Reduction Approaches

Previously, we presented a deep discussion about the main features of the PAPR reduction approaches based on ML in the literature. However, it is essential to discuss the main limitations of the existing solutions to identify possible open research topics. Therefore, we next present a discussion over the main limitations of each previously cited work categorized by the ML technique considered.

#### 4.3.1. ANN [135,141,142,143,144,145,146]

The work in [141] has a couple of important missing points. First, it does not compare how complex the methods are. Second, to understand the BER better, it is crucial to compare it with the original signal and the methods used in the paper. Unfortunately, the paper does not do this. It is unclear if reducing the PAPR keeps the original BER or if the system loses performance in terms of the BER. This missing information makes it hard to fully grasp how reducing the PAPR affects the system performance. In [142], when analyzing the CCDF curves of the PAPR and the symbol power, it is possible to note that while the ACE method decreased the PAPR, it came at the expense of increased signal power. On the other hand, NN-ACE achieved a lower PAPR but at the cost of higher power consumption. Notably, when NN-ACE was implemented, there was a degradation in the overall BER performance. The simulation results in [143] suggest seamless integration into typical Visible Light Communication OFDM (VLC-OFDM) systems, offering favorable PAPR and BER performance even in the presence of ISI in the diffuse optical wireless channel. While acknowledging limitations in real-world implementations and variations in system conditions not fully captured in the simulations, the study provides valuable insights into addressing challenges in indoor VLC-OFDM systems. The work in [144] did not consider the computational complexity of the system, and this information is crucial for evaluating if reduction of the PAPR is applicable or not.

In addition, the authors of [145] mentioned that the proposed method can be used in MIMO scenarios, but they did not provide specific details about the challenges or limitations it might encounter in such situations. The impact on the actual performance of the system in practical MIMO set-ups was not discussed. While the paper notes an improvement in the BER with a reduced PAPR, it does not delve into the trade-off between reducing the PAPR and its potential impact on other important performance metrics. There could be instances where optimizing for one parameter adversely affects another. The paper also does not address how the proposed method reacts to changes in channel conditions or variations in transmission environments. In real-world applications, the robustness of the method to dynamic channel conditions is crucial, and this aspect was not covered. The work in [135] has two notable shortcomings. First, it does not provide a comparison of computational complexity, a crucial aspect that remains unexplored. Second, although the paper includes a table comparing BER and PAPR loss, there is a lack of a detailed explanation in the text. This omission makes it challenging for readers to comprehend whether the BER after PAPR reduction aligned with the original BER, a vital consideration in evaluating the effectiveness of PAPR reduction methods. The authors of [146] recognized certain limitations and emphasized the need for further research, especially in the area of developing faster training methods. This is particularly important for larger OFDM systems that have increased subcarriers and modulation orders. The acknowledgment highlights the awareness of existing constraints and directs attention toward addressing these challenges through advancements in training methodologies.

#### 4.3.2. DL [147,148,149,150,151,152,153,154]

The suggested NN in [147], incorporating specialized domain knowledge, effectively minimizes issues related to LED nonlinearity, nonlinear distortions, and ISI. Additionally, it remains compatible with CP removal. The NN handles interference arising from multipath channels, contributing to an overall enhancement in performance. TRNet in [148] has a substantial training burden, indicating a considerable amount of computational effort and time needed during the training phase. Despite acknowledging this training burden, it is highlighted that the execution time of TRNet is significantly lower than that of the TR technique when utilizing the trained model. The comparison mainly concentrates on the execution time and time complexity rather than the training time. Although TRNet demonstrates efficiency during execution, the acknowledgment of the training burden underscores its practicality in scenarios involving the processing of a large number of symbols. The proposed algorithm in [149] demonstrated better learning capabilities when compared with traditional algorithms, even though its PAPR performance fell short of PRnet. The observation that excessive training data and too many trainable parameters can result in redundancy implies a potential risk of overfitting. Notably, the algorithm demonstrated subpar performance when the number of peak subcarriers surpassed a specific value.

The limitation of the scheme in [150] is that it is described as PAPR-unaware, indicating it does not consider the PAPR in its optimization. This could be problematic in situations where the PAPR is crucial, and not factoring it in might lead to suboptimal performance. Furthermore, the paper offers limited details about the proposed PAPR-aware scheme, making it difficult to evaluate its complexity, feasibility for implementation, and potential drawbacks. This lack of information hinders a comprehensive understanding of the proposed scheme’s characteristics and its effectiveness in addressing PAPR-related challenges. When employing advanced schemes like LACO-OFDM in [151], the transceiver system experienced heightened complexity. This complexity escalation can result in significant nonlinear distortion of the signal, primarily due to the high PAPR of the transmitted signal. Improving PAPR reduction in [152] was achieved by increasing the number of reserved subcarriers. Nevertheless, this performance enhancement reached a saturation point due to a constraint on the transmit power. As more subcarriers were reserved, there was an associated increase in power requirements, potentially leading to the generation of additional side lobes. In [153], the authors suggested that the proposed TT-RDNN method offers advantages in terms of a deterministic PAPR, performance compared to certain existing methods, and the consideration of spectral mask requirements. However, it is important to consider trade-offs, such as spectral efficiency and potential nonlinear distortion in systems where the PAPR exceeds the IBO. The results in [154] indicate that the proposed method allows fine control over the trade-off between the information rate, ACLR, and PAPR. On the Additive white Gaussian noise (AWGN) channel, the learned waveforms achieved competitive rates with conventional waveforms but exhibited significantly lower ACLRs. Lower PAPRs were also possible, though at the cost of a rate loss. The method appears to offer advantages in terms of fine-tuning trade-offs in communication system design, but there are challenges and areas for further exploration, especially for understanding the learned filters and extending the method to different scenarios and technologies.

#### 4.3.3. Other ML Techniques [134,155]

The limitation in [155] is that as the PAPR decreased, it became more challenging to find a better phase vector for achieving a lower PAPR. Consequently, the marginal contribution of the complexity parameter diminished, leading to flattened curves when the parameter was large. However, it is crucial to note that the proposed method did exhibit PAPR performance degradation of 1.16 dB. Finally, in [134], the authors presented a manual optimization of the hyperparameters for each particular situation, a laborious and time-consuming process. In the case of Ridge models, an analytical solution was employed, while in the case of RANSAC, the number of iterations reached a magnitude of $10^{5}$.

### 4.4. Overall ML Technique Limitations

Minimizing the PAPR in wireless communication systems presents a significant challenge, and ML-based approaches have been investigated to address this issue. However, it is crucial to recognize the general limitations associated with these techniques. We highlight below some of these limitations regarding the methods specifically mentioned in this paper:**Computational Complexity:** ML methods often consider complex models with numerous parameters. The requirement of substantial computational resources for training and optimizing these models may limit their applicability in hardware-constrained environments [35].**Interpretability:** Lack of interpretability is a known characteristic of NNs, including ANNs and DNNs. Understanding how these models make specific decisions can be challenging, making it difficult to justify the choices made during the PAPR minimization process [162].**Sensitivity to Training Data:** ML methods such as the ANN and DNN are sensitive to the quality and quantity of the training data. If data are not representative or comprehensive enough, then models may fail to generalize to new scenarios, resulting in suboptimal performance [39].**Limitations of Generalization:** Techniques like MCTS may face difficulties in generalizing to situations outside the training set. Variations in communication channel conditions can compromise the ability of these methods to adapt effectively, leading to less efficient results [163].**Extensive Training Requirements:** Models such as the DUN may require extensive training to achieve satisfactory performance, which may be impractical in scenarios with limited availability of labeled data or when training needs to occur in real time [164].**Theoretical Limitations:** Certain methods, such as RANSAC, may be based on specific theoretical assumptions that may not be fully valid in all practical situations. Variations in channel conditions can compromise the effectiveness of these methods [134].**Trade-off between Accuracy and Computational Time:** PAPR optimization often involves a trade-off between accuracy and processing time. More accurate methods, such as the MCTS and DUN, may require more computational time and are impractical in real-time applications [165].

## 5. Future Directions

In the foreseeable future, addressing numerous challenges within wireless communication systems is paramount, particularly in the context of ML algorithms. Herein, we discuss the challenges of ML in the context of the PAPR.

### 5.1. ML Training

The effectiveness of ML models in optimizing the PAPR hinges on high-quality training data. Acquiring and managing such data poses challenges, leading to the need for protocols in data collection and annotation. To address data scarcity, generative models and data augmentation techniques are essential [45,166].

### 5.2. Generalization across Diverse Scenarios

Generalizing ML models across varied channel conditions, signal types, and evolving wireless standards is a significant challenge. Resilient models that adapt to diverse scenarios require a multi-faceted strategy, including meta-learning, transfer learning, domain adaptation, and the infusion of domain-specific knowledge [45,166].

### 5.3. Resource Constraints in IoT Devices

Implementing ML techniques in resource-constrained devices, such as those in the IoT, introduces challenges. Techniques like model compression, quantization, and hardware acceleration are crucial to balancing computational demands with device constraints [167,168,169].

### 5.4. Real-Time Implementation

Real-time implementation of ML-based PAPR optimization, especially in upcoming wireless standards like 6G, remains a challenge. Low-latency operation demands optimized algorithms and efficient inference techniques, such as model quantization, model parallelism, and hardware accelerators [170,171,172].

### 5.5. Computational Complexity

The computational complexity associated with ML techniques, particularly DL models, presents a central challenge. Specialized hardware and streamlined software frameworks are required to expedite the execution of resource-intensive algorithms, avoiding bottlenecks in real-time wireless communication [173,174].

### 5.6. Adaptability to Dynamic Conditions

Ensuring the adaptability of ML models to rapidly changing wireless environments is an ongoing concern. Techniques like meta-learning, continuous learning, and Reinforcement Learning (RL) are under exploration to make ML models more agile and responsive to dynamic network conditions [175,176].

### 5.7. Energy Efficiency

Energy efficiency is critical for ML-enabled wireless systems, particularly in battery-powered devices. Optimizing energy consumption during training and inference involves techniques such as model sparsity, energy-efficient hardware, and dynamic power management [177,178,179].

### 5.8. Standardization for Seamless Integration

The seamless integration of ML into PAPR optimization solutions relies on standardized interfaces. Open standards and interoperability protocols are essential for the adoption of ML techniques in the wireless industry, encompassing communication, data formats, model interchangeability, and security protocols [180,181].

### 5.9. Robust Wireless Communication Infrastructure

The establishment of a robust wireless communication infrastructure has necessitated addressing prevalent issues such as bit errors, heightened power consumption, and signal distortion. One solution involves deploying sophisticated error correction codes, employing modulation techniques, and implementing innovative signal processing methods to enhance the resilience of communication links against data corruption.

### 5.10. Power Management

Efficient power management is a critical aspect, and it is achieved through the utilization of intelligent algorithms that meticulously balance signal strength and power consumption. Simultaneously, the integration of advanced filters and shaping techniques plays a pivotal role in reducing distortion, ensuring clarity of communication.

## 6. Conclusions

This paper has provided a comprehensive overview of the techniques used in the literature for reducing the PAPR in OFDM systems. We discussed various traditional approaches, novel schemes, and emerging trends in the field. The literature review revealed that conventional techniques, such as PTS, SLM, and clipping, among other techniques, have been widely studied and utilized for PAPR reduction in OFDM. These methods have shown significant improvements in reducing the PAPR. However, they often come with certain drawbacks, such as increased complexity, BER, and distortion. One of the notable advancements in recent years is the integration of ML techniques for PAPR reduction. ML-based approaches have shown great potential in addressing the challenges associated with the PAPR in OFDM systems. These methods leverage the power of data-driven learning and optimization to adaptively and intelligently reduce the PAPR while preserving system performance. 

## Figures and Tables

**Figure 1 sensors-24-01918-f001:**
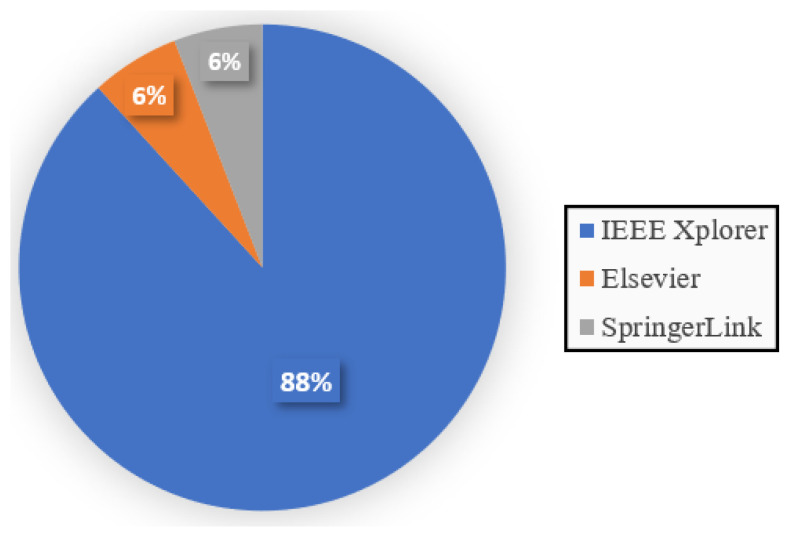
Illustration of the post-selection distribution across the databases.

**Figure 2 sensors-24-01918-f002:**
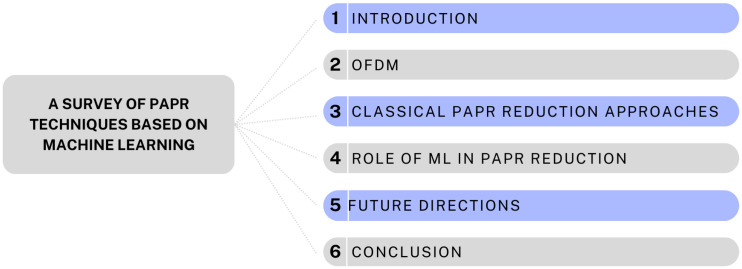
Organization of this survey.

**Figure 3 sensors-24-01918-f003:**
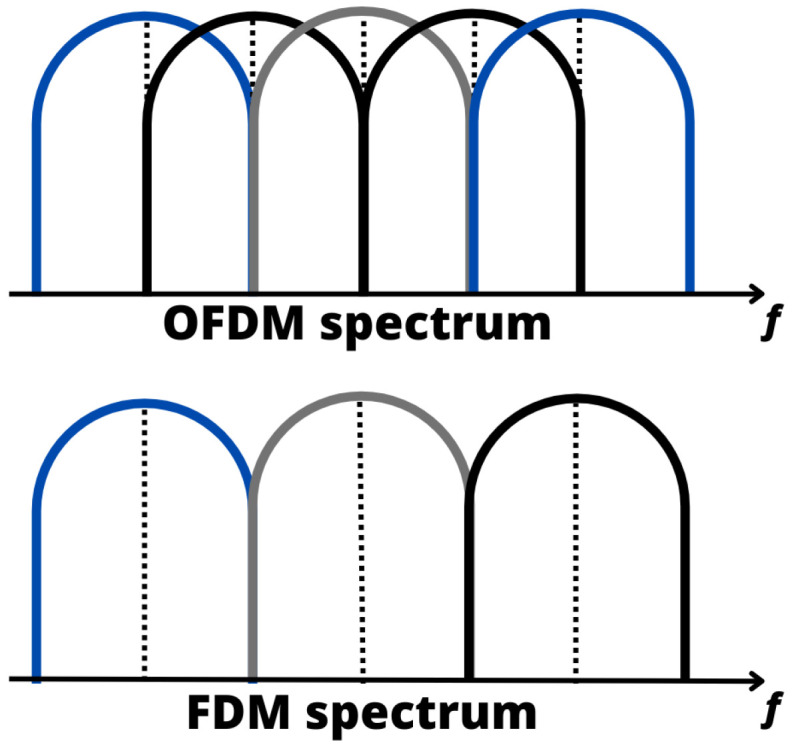
Ideal spectrum illustration for FDM and OFDM techniques.

**Figure 4 sensors-24-01918-f004:**
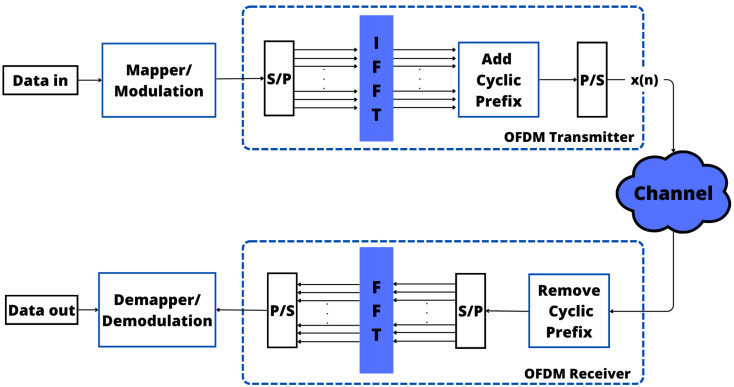
Block diagram of a baseband OFDM system.

**Figure 5 sensors-24-01918-f005:**
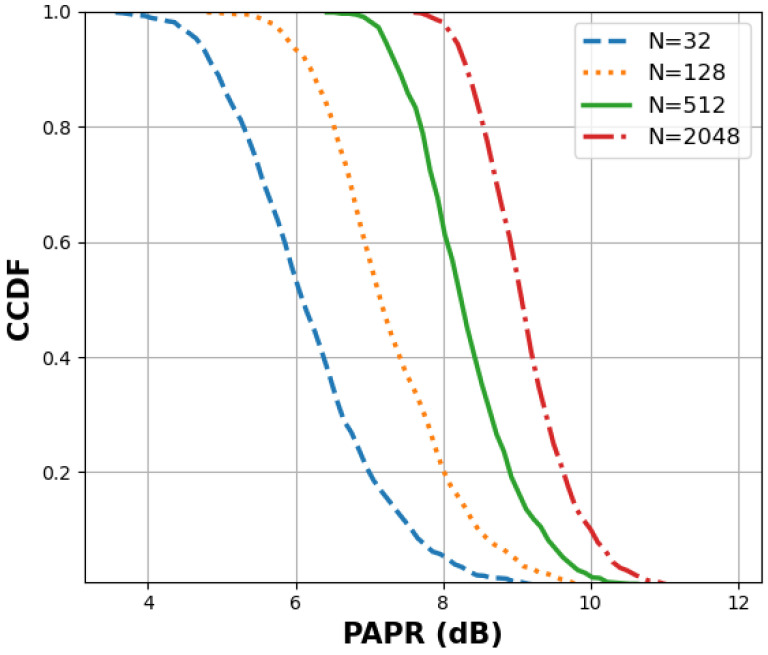
CCDF of the PAPR for 32, 128, 512 and 2048 subcarriers.

**Figure 6 sensors-24-01918-f006:**
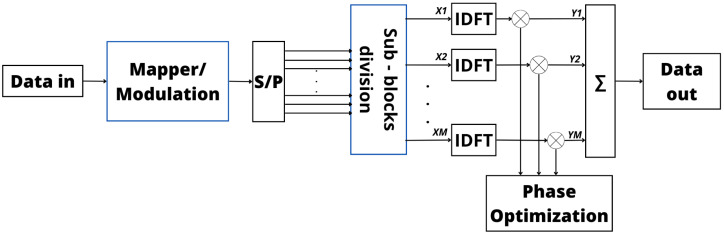
Block diagram of OFDM transmitter with PTS.

**Figure 7 sensors-24-01918-f007:**
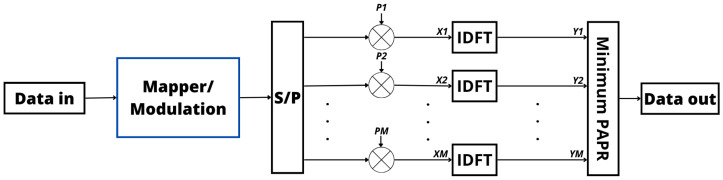
Block diagram of OFDM transmitter with SLM.

**Figure 8 sensors-24-01918-f008:**
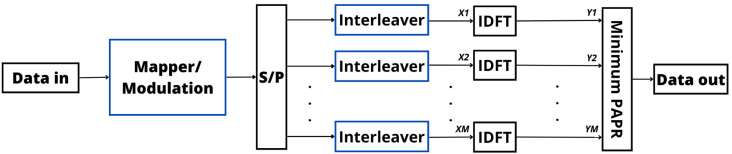
Block diagram of OFDM transmitter with interleaving.

**Figure 9 sensors-24-01918-f009:**
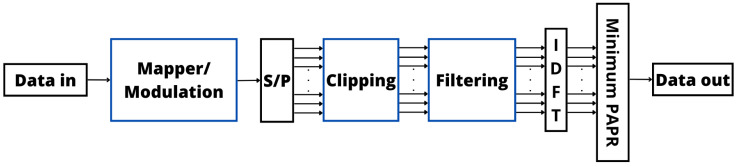
Block diagram of OFDM transmitter with clipping and filtering.

**Figure 10 sensors-24-01918-f010:**
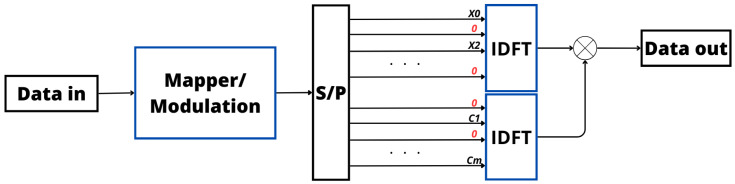
Block diagram of OFDM transmitter for TR.

**Figure 11 sensors-24-01918-f011:**
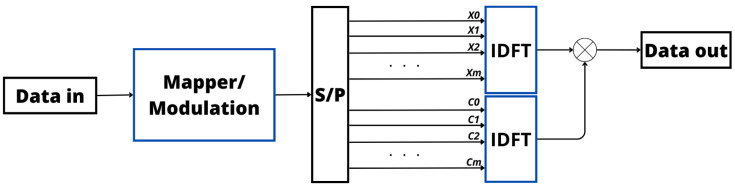
Block diagram of OFDM transmitter for TI.

**Figure 12 sensors-24-01918-f012:**
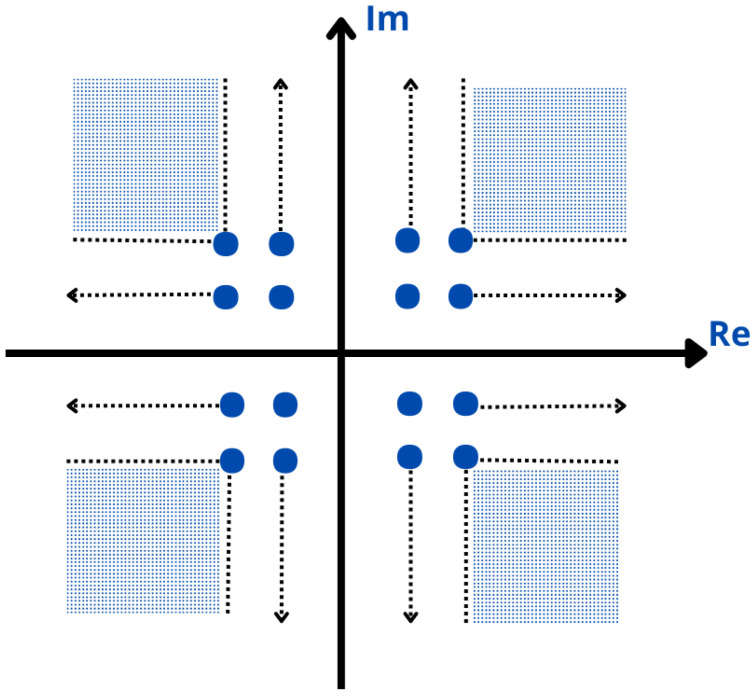
ACE constellation for 16 QAM.

**Figure 13 sensors-24-01918-f013:**
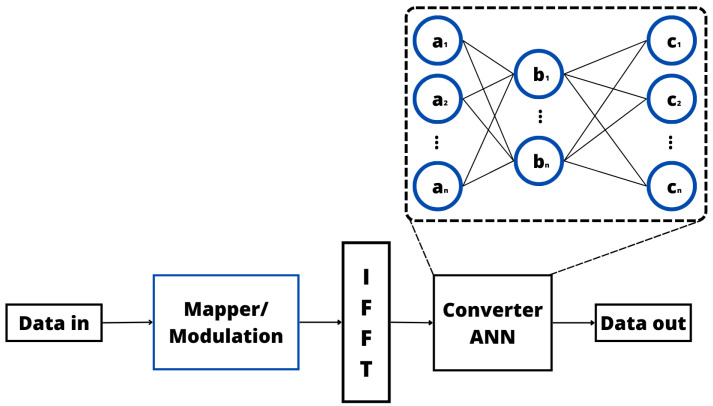
Illustration of the ANN transmitter.

**Figure 14 sensors-24-01918-f014:**
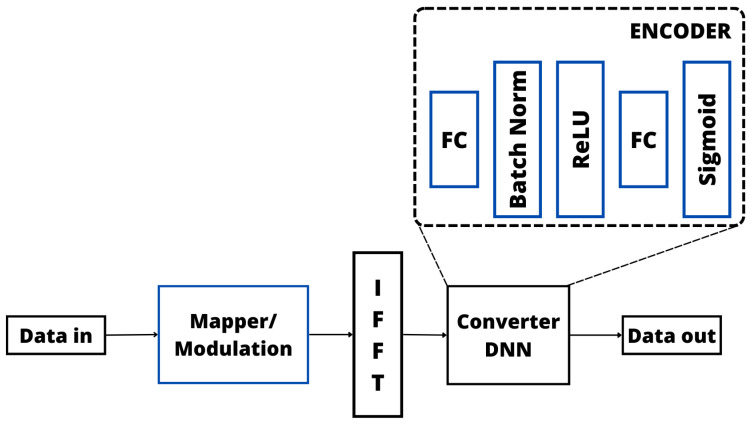
Illustration of the DNN transmitter.

**Figure 15 sensors-24-01918-f015:**
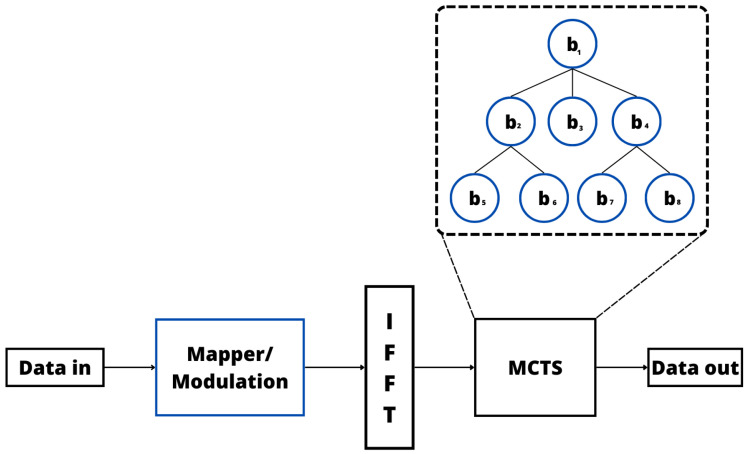
Illustration of the MCTS transmitter.

**Figure 16 sensors-24-01918-f016:**
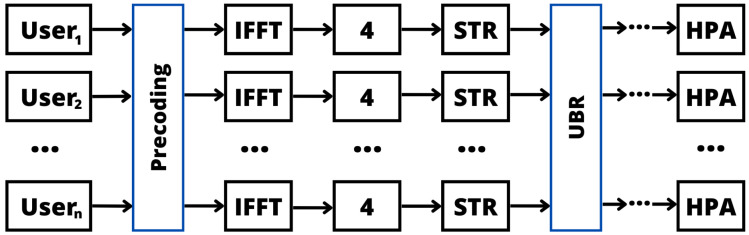
Illustration of the RANSAC transmitter [134].

**Table 1 sensors-24-01918-t001:** Comparison between PAPR reduction techniques.

Technique	BER Increase	Complexity	Side Information	Add Distortion
PTS	no	high	yes	no
SLM	no	high	yes	no
Interleaving	no	high	yes	no
Clipping and Filtering	yes	low	no	yes
TI	no	high	no	no
TR	no	high	yes	no
ACE	no	high	no	no

**Table 2 sensors-24-01918-t002:** Reduction in the PAPR achieved by the ML-based techniques when compared with classical PAPR methods.

Reference	PTS	SLM	Clipping and Filtering	TR	ACE
[141]	4 dB	-	7 dB	-	-
[143]	-	9 dB	6.5 dB	-	-
[147]	3 dB	-	-	-	-
[142]	-	-	-	-	5 dB
[148]	-	-	4.5 dB	7 dB	-
[145]	-	-	3 dB	-	-
[134]	-	-	4.5 dB	-	-
[149]	-	2 dB	-	-	-
[155]	2.5 dB	-	-	-	-
[146]	-	-	2.7 dB	-	-
[152]	-	3 dB	-	-	-
[153]	-	-	4 dB	-	-
[144]	-	7 dB	-	-	-

**Table 3 sensors-24-01918-t003:** Overview of the state of the art of PAPR reduction approaches based on ML (2019–2023).

Year	Reference	ML Technique	Goal
2019	[141]	ANN	Minimize the PAPR.
2019	[143]	ANN	Enhance indoor VLC-OFDM system performance.
2019	[147]	DNN	Enhance BER performance in VLC.
2019	[142]	ANN	Reduce the PAPR of OFDM symbols.
2020	[148]	DNN	Used to generate peak-canceling signal to reduce PAPR.
2020	[145]	ANN	Reduce PAPR in wireless transmitters.
2020	[134]	RANSAC	PAPR Reduction in Massive MIMO
2021	[149]	DNN	PAPR reduction in OFDM system.
2021	[151]	DNN	PAPR reduction using O-OFDMNet.
2021	[155]	MCTS	PAPR Reduction for OFDM signals.
2022	[146]	ANN	Enhance VLC system performance by reducing PAPR.
2022	[152]	DUN	Address PAPR challenges and power constraints.
2022	[135]	ANN	Frequency-selective PAPR reduction.
2022	[153]	DNN	Mitigate high PAPR in multicarrier transmissions.
2022	[154]	DNN	Enhance spectral efficiency and mitigate PAPR issues.
2023	[150]	DNN	Maximize energy efficiency and reduce PAPR.
2023	[144]	ANN	Reduce PAPR of GFDM.

**Table 4 sensors-24-01918-t004:** Contributions and Limitations of the PAPR reduction approaches based on ML (2019–2023).

Reference	Contributions	Limitations
[141]	PAPR is reduced to 6.5 dB compared with conventional methods.	Lack of complexity comparison, unclear if reducing PAPR maintains original BER, uncertainty about system performance impact.
[143]	A 12 dB PAPR reduction and improved BER performance.	Real-world implementations, variations in system conditions not fully captured in simulations.
[147]	Effective LED nonlinearity mitigation and improved BER performance.	The randomness and instability of network training indicates that there may be challenges in the neural network training phase, making it a challenge to obtain more reliable and consistent performance.
[142]	Achieve a lower PAPR and reduce the BER in a nonlinear channel model.	NN-ACE raises power consumption. Degradation in overall BER performance when NN-ACE is implemented.
[148]	Provide better PAPR reduction performance with fewer TRs.	Considerable computational effort and time in the training phase.
[145]	PAPR reduction and BER minimization through joint offline training.	Absence of exploration of trade-offs between PAPR reduction and other performance metrics.
[134]	PAPR reduction based on the classical ML approaches.	Time-consuming optimization of hyperparameters. RANSAC requires a high number of iterations.
[149]	Achieve comparable PAPR performance with low complexity and training costs.	Risk of overfitting due to excessive training data and numerous trainable parameters. Subpar performance is observed when the number of peak subcarriers exceeds a specific value.
[151]	O-OFDMNet optimizes BER and PAPR in soft decision channel decoder.	Heightened complexity in LACO-OFDM results in significant nonlinear distortion, primarily due to the high PAPR of the transmitted signal.
[155]	Reduced complexity, PTS and PAPR performance improvement.	As PAPR decreases, finding an optimal phase vector becomes harder. The proposed method presents a PAPR drop of 1.16 dB.
[146]	PAPR reduction method that effectively improves VLC performance while achieving a balanced compromise with the BER.	High time consumption in the training process.
[152]	Significant PAPR reduction, BER, and reduced computational load.	Performance enhancement reaches saturation point due to a constraint on the transmit power. More reserved subcarriers lead to an increase in power requirements, potentially generating additional side lobes.
[135]	PAPRer offers favorable performance complexity for frequency-selective PAPR reduction.	Lack of comparison of computational complexity.
[153]	PAPR reduction while upholding spectral efficiency and minimizing in-band distortion through the TT-RDNN technique.	Does not consider important trade-offs, such as spectral efficiency and potential nonlinear distortion in systems where the PAPR exceeds the IBO.
[154]	PAPR reductions compared with traditional methods.	Difficulty balancing trade-offs in information rate, ACLR, and PAPR. Challenges in multiplexing and better interpretation of learned filters.
[150]	A DL solution for a 3 dB higher energy efficiency and a 3 dB PAPR reduction compared with conventional methods.	Lack of complexity analysis, feasibility for implementation, and potential drawbacks.
[144]	The PAPR and BER of the GFDM system are minimized considerably.	Lack of computational complexity analysis.

## Data Availability

Data are contained within the article.

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
