# Peer review of "A Survey of PAPR Techniques Based on Machine Learning"

_sensors, 2024, doi:10.3390/s24061918_

Round 1
Reviewer 1 Report
Comments and Suggestions for Authors
1. The paper provides very basic information about FDM and OFDM. What is the motivation behind this work? The contributions of this work are very limited.
2. A review article analyzes & discusses the methods and conclusions in previously published studies. However, in this paper, there is no analysis & discussion of previous studies. The article just provides a summary of previous techniques and a summary of ML-based PAPRP techniques. The authors are advised to read some previously published review articles and rewrite the article accordingly.
3. How figure 4 is obtained? What are the parameters used? Why only the CDF plot is considered?
4. The title of the paper is “A Survey of PAPR Optimization Techniques based on ML for Beyond 5G Systems” whereas there is only one section on ML-based PAPR technique (section 4). Moreover, the section only presents a literature survey. There is no discussion, about the limitations of these works.
Comments on the Quality of English LanguageThe author should throughly proofread the paper.
Author Response
Dear Reviewer,
We thank the Reviewer for his/her generosity and precious time, and for giving us the opportunity to clarify and improve the contributions of our work. Therefore, all the responses to the reviews are in the attached document.
Best Regards
Bianca S. da Silva and co-authors.

Reviewer 2 Report
Comments and Suggestions for Authors
This paper is a survey paper on PAPR reduction. However, this requires significant revision.
Most of the paper discusses the classical PAPR reduction techniques, which have been available widely in many resources for a long time. Only the last few pages of the paper discuss the ML-based PAPR reduction.
The authors elaborated on the procedure of classical PAPR reduction schemes but not ML-based schemes, which is essential.
Even though the title is PAPR optimization, there is no discussion on the optimization part. Is the term optimization relevant?
The title talks about 5G. However, there is no discussion about 5G-NR and its specifications.
A detailed discussion on the complexity of ML/DL/ANN algorithms is required.
A discussion on which is preferable among ML/DL/ANN is missing.
The authors should highlight their contributions. How is this review article better than the other review articles on PAPR reduction?
Even though this is a review paper, a few existing work results are essential.
The amount of PAPR reduced in every scheme must be tabulated.
Comments on the Quality of English LanguageMinor English corrections are required
Author Response
Dear Reviewer,
We thank the Reviewer for his/her generosity and precious time, and for giving us the opportunity to clarify and improve the contributions of our work. Therefore, all the responses to the reviews are in the attached document.
Best Regards,
Bianca S. da Silva and co-authors.

Reviewer 3 Report
Comments and Suggestions for Authors
Please see attached file

Minor grammar errors
Author Response

(The authors gave the same response as above.)

Round 2
Reviewer 1 Report
Comments and Suggestions for Authors
1. The contributions of this work are very limited. Section 2 is just a summary of OFDM and Section 3 is Classical PAPR Reduction Approaches. Both sections present very basic information.
2. Role of ML in PAPR Reduction presented in seciton 4. The section presents a summary of ML-based techniques. There is no analysis of previously published work.
3. Figure 4 presents a CCDF plot of the PAPR for different subcarriers. However, there is no plot for ML-based techniques which shows that ML-based techniques can perform better as compared to traditional approaches.
Comments on the Quality of English LanguageThe author should thoroughly proof read the paper.
Author Response
Dear Reviewer,
We thank the Reviewer for your generosity and precious time, and for giving us the opportunity to clarify and improve the contributions of our work. Therefore, all the responses to the reviews are in the attached document.
Best Regards
Bianca S. da Silva and co-authors.

Reviewer 2 Report
Comments and Suggestions for Authors
The authors addressed some of my comments. But the work needs further revision for considering in a high impact journal.
The authors presented detailed methodology, block diagrams for the classical PAPR reduction schemes. A similar approach has to be followed for the machine learning-based methods. Instead, authors have gathered various existing works and presented them. For example, deep learning means the corresponding blocks, mathematical formulations, features, etc. must be added. A detailed analysis is required to understand how the architecture reduces PAPR. The superiority of this approach over conventional methods in terms of metrics and computational complexity should also be examined.
Comments on the Quality of English LanguageModerate English corrections required
Author Response

(The authors gave the same response as above.)

Reviewer 3 Report
Comments and Suggestions for Authors
English is okay
Author Response

(The authors gave the same response as above.)

Round 3
Reviewer 1 Report
Comments and Suggestions for Authors
The authors have addressed the previous round comments. Only one minor suggestion move the "Acronyms and Symbols" to the start of the paper before the introduction or on the second page.
Reviewer 2 Report
Comments and Suggestions for Authors
The authors addressed all my comments.
Reviewer 3 Report
Comments and Suggestions for Authors
All concerns has been addressed by authors